# Neoadjuvant Statistical Algorithm to Predict Individual Risk of Relapse in Patients with Resected Liver Metastases from Colorectal Cancer

**DOI:** 10.3390/biomedicines12081859

**Published:** 2024-08-15

**Authors:** Ángel Vizcay Atienza, Olast Arrizibita Iriarte, Oskitz Ruiz Sarrias, Teresa Zumárraga Lizundia, Onintza Sayar Beristain, Ana Ezponda Casajús, Laura Álvarez Gigli, Fernando Rotellar Sastre, Ignacio Matos García, Javier Rodríguez Rodríguez

**Affiliations:** 1Department of Medical Oncology, Clínica Universidad de Navarra, 31008 Pamplona, Spain; avizcay@unav.es (Á.V.A.); tzumarraga@unav.es (T.Z.L.); 2Department of Mathematics and Statistic, NNBi, 31110 Noain, Spain; olast.arrizibita@nnbi.es (O.A.I.); oskitz.ruiz@nnbi.es (O.R.S.); onintza.sayar@nnbi.es (O.S.B.); 3Department of Radiology, Clínica Universidad de Navarra, 31008 Pamplona, Spain; aezponda@unav.es; 4Department of Pathology, Clínica Universidad de Navarra, 31008 Pamplona, Spain; lagigli@unav.es; 5Department of HPB Surgery, Clínica Universidad de Navarra, 31008 Pamplona, Spain; frotellar@unav.es; 6Department of Medical Oncology, Clínica Universidad de Navarra, 28027 Madrid, Spain; imatos@unav.es

**Keywords:** colorectal cancer (CRC), liver metastases (LM), statistical algorithms, gradient boosting machine (GBM), synthetic data

## Abstract

(1) Background: Liver metastases (LM) are the leading cause of death in colorectal cancer (CRC) patients. Despite advancements, relapse rates remain high and current prognostic nomograms lack accuracy. Our objective is to develop an interpretable neoadjuvant algorithm based on mathematical models to accurately predict individual risk, ensuring mathematical transparency and auditability. (2) Methods: We retrospectively evaluated 86 CRC patients with LM treated with neoadjuvant systemic therapy followed by complete surgical resection. A comprehensive analysis of 155 individual patient variables was performed. Logistic regression (LR) was utilized to develop the predictive model for relapse risk through significance testing and ANOVA analysis. Due to data limitations, gradient boosting machine (GBM) and synthetic data were also used. (3) Results: The model was based on data from 74 patients (12 were excluded). After a median follow-up of 58 months, 5-year relapse-free survival (RFS) rate was 33% and 5-year overall survival (OS) rate was 60.7%. Fifteen key variables were used to train the GBM model, which showed promising accuracy (0.82), sensitivity (0.59), and specificity (0.96) in predicting relapse. Similar results were obtained when external validation was performed as well. (4) Conclusions: This model offers an alternative for predicting individual relapse risk, aiding in personalized adjuvant therapy and follow-up strategies.

## 1. Introduction

CRC poses a significant epidemiological and health challenge due to its high incidence, ranking second among women (after breast cancer) and men (after prostate cancer), as well as its considerable mortality, ranking second after lung cancer. At the time of diagnosis, approximately 20% of patients already have LM, and over 50% will develop it during their illness. The presence of LM in CRC is not only common but also leads to complications that are the primary cause of death in these patients [1,2,3,4,5,6].

The management of patients with metastatic CRC primarily follows international guidelines [7,8]. Within this context, patients with upfront resectable or potentially resectable LM from CRC form a diverse group in terms of prognosis, with reported 5-year overall survival rates ranging between 35% and 60%. Despite this prognostic variability, clinical management is generally uniform and revolves around a multidisciplinary approach combining systemic treatment and surgical resection, albeit with minor variations depending on the specific clinical scenario [9,10,11,12]. Moreover, the criteria for determining the resectability of LM are broad, necessitating the involvement of multidisciplinary committees in the decision-making process [13,14,15,16,17].

It is desirable to identify patients with varying probabilities of relapse who may potentially benefit from novel adjuvant treatment approaches, reconsideration of surgical timing, or more intensive follow-up. In clinical practice, several nomograms are utilized for this purpose [18,19,20,21,22]. These mathematical tools consider various clinical, analytical, and radiological parameters to define patient subgroups with different risks of relapse after surgery. However, the accuracy of these nomograms remains suboptimal [23,24,25,26].

The objective of this investigation is to assess the individual risk of relapse following LM resection using interpretable statistical algorithms that combine clinical, analytical, radiological, molecular, and pharmacokinetic parameters [27,28,29,30] to provide relevant information for tailoring adjuvant treatments and follow-up protocols.

## 2. Materials and Methods

### 2.1. Patient Eligibility

We retrospectively analyzed a homogenous cohort of 86 patients with potentially resectable LM of CRC, with an Eastern Cooperative Oncology Group performance-status score of 0 or 1 [31], and an adequate hematological, kidney and liver functions, who underwent complete surgical resection after systemic preoperative treatment. A multidisciplinary tumor board, comprising medical oncologists, hepatobiliary surgeons, radiologists, pathologists, and radiation oncologists, evaluated each case. The criteria for surgical resectability were established individually before the initiation of neoadjuvant treatment. Initial evaluation included: clinical examination, laboratory tests (including serum CEA tumor marker levels), and a thoracoabdominal CT scan to define the extent of disease.

### 2.2. Neoadjuvant Therapy

The neoadjuvant treatment regimen consisted of standard first-line chemotherapy administered for a duration of 2–4 months. The chemotherapy options included FOLFOXIRI (5-Fluorouracil, Oxaliplatin, and Irinotecan), XELOXIRI (Capecitabine, Oxaliplatin, and Irinotecan), FOLFOX (5-Fluorouracil and Oxaliplatin), or XELOX (Capecitabine and Oxaliplatin). Prior to each chemotherapy cycle, all patients underwent routine examinations, which included physical examination, blood tests, and assessment of treatment-related adverse effects. 

During treatment, each patient underwent at least two radiological CT scans. The first scan was performed before the initiation of chemotherapy as a baseline assessment, and subsequent scans were conducted during chemotherapy and/or before the surgery. These radiological examinations were evaluated separately by a radiologist using the RECIST 1.1 criteria. 

Surgery was scheduled 4–6 weeks after the end of the neoadjuvant protocol, once progressive disease was ruled out by preoperative thoracoabdominal CT ± liver MRI. The specific surgical approach, as well as the need for portal embolization and/or intraoperative radiofrequency ablation, was determined by the hepatobiliary surgeons to ensure R0 resection. 

Although it is not standard practice, the administration of adjuvant treatment was left to the clinician criteria considering pathological findings, patient characteristics, and any associated comorbidities. Patients were initially monitored at intervals of every 3–4 months for the first two years, followed by evaluations every 6 months during years 3 and 4, and subsequently on an annual basis. The surveillance protocol encompassed physical examinations, laboratory assessments, and thoracoabdominal CT scans (Figure 1).

### 2.3. Clinical Information

The following variables were retrospectively analyzed for each patient: gender, age, performance status, primary tumor location, clinical stage (pTNM), type of chemotherapy, number of cycles administered, dosage, dose intensity (mg/m^2^/week), toxicity profile, radiological response on the basis of RECIST (version 1.1), endogenous pyrimidine levels, area under the curve of 5-fluorouracil, type of surgery, pathological stage (pTNM), number of resected lymph nodes, presence of vascular and perineural invasion, tumor regression grade, surgical margins, date and type of disease relapse (local and/or distant), and utilization of subsequent treatments. Additionally, liver and renal function, complete blood count, coagulation profile, tumor markers (with emphasis on CEA), as well as the ratios of neutrophils and platelets to lymphocytes (NLR: neutrophil–lymphocyte ratio and NPR: neutrophil–platelet ratio) were analyzed from the initiation of neoadjuvant treatment to oncologic surgery.

### 2.4. Statistical Analysis

The statistical analyses were initially carried out with the statistical software R (Rstudio, R version 4.3.3 29 February 2024), by the Department of Mathematics and Statistic, NNBi. To analyze quantitative variables that were measured over time, new variables were generated for each of them, calculating the maximum, minimum, standard deviation, mean, median, and difference between the first and last values. Therefore, twelve variables were obtained from each of the originally measured. Statistical analysis was divided into two steps: (1) clean-up and general analysis (descriptive, outliers, missing values, and basic assumptions), (2) correspondence analysis. 

After a rigorous statistical analysis, 12 patients were excluded from the study based on specific criteria: 3 patients lacked information on the dependent variable and 9 patients were considered outliers. The exclusion of outliers is recommended to mitigate their potential influence on the training of the predictive model. The identification of outliers was conducted using the Mahalanobis distance method [32,33]. Notably, individuals exceeding the established threshold (*p* > 0.001), determined by the distribution of distances, were considered as outliers. This approach enhances the efficacy of the model by mitigating the variability inherent in variables that substantially diverge from the dataset. Consequently, the predictive model was developed using data from a cohort of 74 patients.

In this study, correspondence analysis [34] was performed to compare the ECOGT variable (ECOG scale after neoadjuvant therapy) with the relapse variable (Figure 2). The analysis revealed that ECOGT Categories 1 and 2 were closely associated with Category 1 of the relapse variable. Conversely, ECOGT Category 0 showed a strong alignment with Category 0 of the relapse variable. Based on these findings, the decision was made to reduce the dimensionality of the ECOG variable to two options: 1- ECOGT_0 Yes and 2- ECOGT_0 No. This simplification allows for a more concise representation of the ECOG variable, emphasizing its significant relationship with the relapse variable.

OS was defined as the duration from the initiation of neoadjuvant treatment until either death from any cause or the last recorded live contact. RFS was calculated from the moment of the surgical intervention until the occurrence of disease relapse (local and/or distant), death from any cause, or the last contact without recurrence. OS and RFS were estimated using the Kaplan–Meier method. The neoadjuvant period was defined as the duration from the initiation of neoadjuvant chemotherapy treatment to the surgical date.

This retrospective observational study received approval from the Clinical Research and Ethics Committee of Navarra (registration number 2021.060).

#### 2.4.1. Variable Creation

In our research, we employed statistical techniques to generate new variables from the original data, with the aim of exploring potential relationships among them and to improve the accuracy of the predictive model. These new variables encompassed various measures, including the difference between the first and last recorded values (denoted as “esti_difer”), the minimum, maximum, standard deviation, and mean of each variable.

#### 2.4.2. Logistic Regression

In this research, we selected the LR method to develop a predictive model due to its interpretability, which is crucial in medical practice. Interpretable methods provide a clear understanding of model performance, allowing for verification of predictions and comprehension of the underlying reasoning [35]. After conducting statistical analysis, a valid dataset was obtained to initiate the process of building the predictive model. Initially, we identified variables with significant statistical significance in predicting relapse through univariate tests using LR. Subsequently, we designed a series of models by iteratively adding and removing predictor variables to achieve the best model fit with the lowest complexity.

To determine the optimal model, we performed ANOVA tests on models with different numbers of variables, prioritizing those with the highest univariate statistical significance. This approach enabled us to compare the predictive performance of each model and assess whether adding a new variable significantly improved accuracy. The overall fit of each model was evaluated using the likelihood ratio chi-square statistic, allowing for us to determine whether a model was significantly better than the previous one [36].

By employing this iterative model-building approach and comparing models with varying numbers of variables, we ensured that the final model included only the most relevant predictor variables, thus avoiding overfitting to the training data. The resulting predictive model underwent validation using stratified k-fold cross-validation, where the data were randomly divided into k subsets (in this case, five) while maintaining a constant proportion of patients in each class within each subset. Specifically, we aimed to maintain the same proportion of patients who experienced relapse and those who did not, as observed in the overall dataset (64% relapse and 36% no relapse). During each iteration of k-fold cross-validation, k-1 subsets were used for training the algorithm, and the remaining subset was used for testing the model. This process was repeated k times, covering all possible combinations between the subsets, and the resulting arithmetic mean was used as the final estimate of model performance. Additionally, after cross-validation with the chosen final model, internal and external validation processes were conducted using a fresh, independent dataset consisting of 23 and 24 patients, respectively, who met the same inclusion criteria as the original sample.

Given the anticipated limited information, GBM was considered to improve model accuracy and robustness. Additionally, synthetic data were utilized to enhance model training due to the restricted number of patients and insufficient representation of certain subgroups.

#### 2.4.3. Gradient Boosting Machine (GBM)

In this research, we selected the GBM artificial intelligence method, renowned for its ability to construct highly accurate predictive models from complex datasets. GBM operates by iteratively combining multiple simple models such as decision trees to enhance the accuracy of the final model. This approach stems from the concept of boosting, wherein additional models are introduced to correct the errors of preceding models, thereby producing a more robust and accurate overall model. A distinctive feature of GBM is its capacity to explore a wide range of potential functions to identify the one that best fits the training data [37,38].

#### 2.4.4. Synthetic Data Generation

Due to the limited training data, the database’s record count was increased to achieve a more robust model training. Synthetic data generation techniques utilizing Bayesian Networks (BNs) and the method of differential privacy (DP) were employed. By integrating these methods, the dataset was expanded while ensuring data anonymization. This process not only increased the size of the dataset but also improved the diversity and representativeness of the different subpopulations within it. This process was facilitated by implementing a synthetic data generation algorithm based on BN, specifically DataSynthesizer [39,40].

#### 2.4.5. SHAPs (Shapley Additive Explanations)

SHAPs is a method based on cooperative game theory and used to increase transparency and interpretability of machine learning models. SHAPs values analysis enabled the identification of variables exerting the greatest influence and their impact on the predictive variable. SHAPs values offer precise insights into the contribution of each variable to model predictions, elucidating their individual roles in shaping outcomes.

## 3. Results

### 3.1. Clinical Data

#### 3.1.1. Patient Characteristics

The baseline characteristics of the 74 CRC patients included in the retrospective analysis are summarized in Table 1. The median age at diagnosis was 59 years (range 28 to 79), with a male-to-female ratio of 47 to 27. Among the patients, 59 (79.7%) had ECOG performance status of zero, while 15 (20.2%) had a status of one. In terms of tumor location, 71.6% of patients had primary tumors in the colon, while 28.4% had rectal tumors.

None of the patients with metachronous disease had received prior adjuvant chemotherapy or radiotherapy; eight patients had a low-risk stage II colon cancer and subsequently were not candidates to systemic therapy. The remaining four patients had stage III colon cancer, but they refused adjuvant chemotherapy. Among those with rectal tumors and synchronous liver metastases, the majority (73%) underwent radiation therapy as part of their neoadjuvant treatment. The median number of liver metastases was two, ranging from one to sevent.

Neoadjuvant polychemotherapy included triplet therapy with oxaliplatin, irinotecan and fluoropyrimidines (n = 54; 74%), or doublets with oxaliplatin and fluoropyrimidines (n = 18, 26%). The median number of preoperative cycles was four (range one to eleven). The time interval from chemotherapy to surgery ranged from 30 to 100 days. All patients achieved complete resection (R0).

Following the surgical intervention, 55 patients (76.4%) underwent adjuvant treatment, with most of them (80%) receiving the same protocol as the one used in the neoadjuvant phase. The median number of adjuvant cycles administered was three, ranging from zero to six. 

#### 3.1.2. Surgical Outcome and Pathological Results

All patients underwent surgery for both colorectal and hepatic tumor; 17.6% underwent simultaneous surgery for the primary tumor and the liver metastases, while the remaining 82.4% had separate surgical procedures. Liver surgical approaches included atypical resections, segmentectomies, and hepatectomies. Portal embolization and liver radiofrequency were performed in 10.8% (8/74) and 25.7% (19/74) of the patients, respectively.

#### 3.1.3. Patients’ Long-Term Outcome

The 5-year RFS rate was 33%, and the 5-year OS rate was 60.7% after a median follow-up of 58.3 months (range 10 to 218 months). At 12, 24, 36, and 48 months, the RFS rates were 70%, 50%, 38%, and 34%, respectively. The OS rates at the same time intervals were 98.6%, 86.1%, and 79.2% and 73.7%, respectively (Figure 3). 

A total of 48 patients (64.9%) relapsed, with most relapses occurring in the liver (22/48, or 45.8%) and lungs (8/48, or 16.7%), respectively. Among liver and lung relapses, 63.3% (19/30) were surgically resected. Additionally, 19.6% (9/48) of patients experienced relapses involving at least two different organs. Every patient who relapsed received second-line therapy.

#### 3.1.4. Synthetic Patient Data

A total of 850 records corresponding to simulated individuals were generated from an original database. These records were meticulously crafted to reflect realistic patterns and demographic characteristics representative of the original population, with no statistically significant differences observed.

### 3.2. Predictive Population Model

#### 3.2.1. Model Development

Prognostic features associated with the risk of relapse in CRC patients were collected and are presented in Table 2. After applying statistical techniques combined with artificial intelligence data (GBM, synthetic data, SHAP values), the most influential variables for the development of the predictive model were as follows: mean and maximum of AST (aspartate aminotransferase), mean and minimum of creatinine, asthenia, liver metastases regression grade (TRG), origin of the primary tumor, ECOG performance status after neoadjuvant treatment (ECOGT), platelet standard deviation, minimum seconds to coagulate blood, the differential estimator of VPM (mean platelet volume), platelets, PTC (prothrombin time concentration), eosinophils, and CHCM (mean corpuscular hemoglobin concentration).

#### 3.2.2. Model Interpretation

The SHAP plot presented in Figure 4, representing the top 10 most influential features, provides a comprehensive insight into their contribution to our GBM model. This plot is pivotal in understanding how each feature influences the model’s predictions, either positively or negatively. The accuracy, sensitivity, and specificity scores (value and 95% CI) for the model were 0.83 95% CI: (0.76, 0.88), 0.59 95% CI: (0.47, 0.71), and 0.96 95% CI: (0.93, 0.96).

The mean of AST shows a significant positive contribution to the model’s predictions, increasing the prediction probability by approximately 20 units. This suggests a strong association between higher values of AST mean and the predicted outcome.

Conversely, the VPM differential estimator has a modest negative impact, reducing the prediction probability by around 0.4 units. This indicates that as the VPM differential estimator increases, the likelihood of the predicted outcome slightly decreases.

On the other hand, the standard deviation of platelets significantly reduces the prediction probability by approximately 24 units, suggesting a strong inverse relationship between the standard deviation of platelets and the predicted outcome.

Another critical feature, minimum seconds to blood coagulation, decreases the prediction probability by about 30 units, highlighting its substantial negative influence on the model’s output.

Minor contributions are observed from features such as PTC (prothrombin time concentration) differential estimator and eosinophils differential estimator, which decrease the prediction probability by approximately 0.1 units. Although their impacts are slight, they do contribute to the overall predictive mechanism of the model.

Additionally, tumor regression grade has a positive contribution of 1 unit, indicating a beneficial effect on prediction probability. Meanwhile, asthenia and eosinophils differential estimator show negligible impacts, with contributions around 0 and 0.5 units, respectively.

Notably, the platelet differential estimator significantly decreases the prediction probability by approximately 63 units, underscoring its substantial negative influence.

#### 3.2.3. Internal Validation of the Model

In our internal model validation, we utilized a sample of 23 patients with similar clinical characteristics (Table 3). Among the 23 patients, 20 experienced relapses. It is important to note that the relapse rate observed in this study may not reflect actual rates and should be considered when interpreting the results.

Table 4 provides a summary of the descriptive analysis of the variables in the internal validation model dataset. As shown in Table 5, the model’s predicted outcomes align with the actual outcomes in 19 out of 23 patients, resulting in an individual-level predictive accuracy of 78%. The sensitivity and specificity of the model are 25% and 89%, respectively. The low sensitivity is primarily due to the small number of instances where “no improvement” is predicted, which is consistent with the actual data. Consequently, the model is highly sensitive to low values in this regard.

#### 3.2.4. External Validation of the Model

In our external validation, we utilized a sample of 24 patients with similar clinical characteristics. Among the 24 patients, 18 experienced relapses. It is imperative to acknowledge that the relapse rate observed in this study may not accurately represent real-world occurrences and should be taken into consideration when interpreting the findings.

Table 6 provides a summary of the descriptive analysis of the variables in the external validation model dataset. As shown in the confusion matrix in Table 7, the model’s predicted outcomes aligned with the actual outcomes in 18 out of 24 patients, resulting in an individual-level predictive accuracy of 71%. The sensitivity and specificity of the model were 50% and 78%, respectively. The sensitivity is not ideal as it does not exceed the minimum threshold of 70%. However, considering that the amount of data we have is not substantial, the model is highly sensitive to the data due to the small sample size.

## 4. Discussion 

Despite advancements in the management of CRC patients with LM, the relapse rate continues to be significant even with multimodal treatment strategies. Although prognostic nomograms exist in this context, the selection of neoadjuvant therapy and the optimal follow-up program primarily relies on the discretion of the attending physician. 

The most widely used nomogram to predict recurrence after resection of LM from CRC is the clinical risk score (CRS) system [42] based on two imaging factors (number and size of metastases) and three oncological parameters (disease-free interval, CEA, node-positive primary tumor), and the aim of the score is to predict the long-term outcome of patients operated on for hepatic metastases from colorectal cancer. Other available tools take into consideration distinct factors (such as pathological staging of the primary tumor, tumor burden, synchronicity, preoperative CEA levels, lymph node engagement, surgical margin status, tumor regression degree, etc.) to delineate cohorts of patients with varying levels of post-surgical recurrence risk. These scales are primarily designed to assist in the timing of surgical intervention and have limitations such as heterogeneity in clinical assessment and limited individual risk prediction [43,44,45]. 

Relevant variables according to our model include mean and maximum of AST, mean and minimum of creatinine, asthenia, TRG, origin of the primary tumor, ECOGT, platelet standard deviation, minimum seconds to coagulate blood and the differential estimator of VPM, platelets, PTC, eosinophils and CHCM. Each of these variables significantly impacts the model’s predictions, either increasing or decreasing the probability of relapse. Our model achieved accuracy, sensitivity, and specificity scores of 0.82, 0.59, and 0.96, respectively. In the validation cohort, precision and specificity scores were 0.78 and 0.89, respectively. However, the sensitivity score of 0.25 was lower than anticipated in the internal validation, indicating room for improvement in accurately predicting negative outcomes. In the external validation, there was some improvement in sensitivity, though it remains suboptimal. Furthermore, the external validation data are consistent with the predictive model generated and the values obtained from the internal validation, indicating the robustness of the algorithm developed. The results obtained for accuracy, sensitivity, and specificity are 0.71, 0.5, and 0.78, respectively.

The relationship between renal function and CRC should be considered in the context of complex reciprocity: impaired renal function is a risk factor for CRC and CRC may influence renal capacity in these patients. In our model, we determined that creatinine levels during neoadjuvant treatment could serve as a prognostic factor. Given the strong association between creatinine and muscle mass in these conditions, sarcopenia appears to be the most plausible explanation for low creatinine levels. However, these mechanisms are highly complex and require further exploration. Some studies have shown a correlation between low creatinine levels during hospitalization and an increased risk of death. Regardless, a patient with low creatinine levels should be considered to have sarcopenia, and it is recommended to conduct nutritional assessment and reevaluate their dietary habits and physical activity levels [46,47].

We found a consistent relationship between the time it takes for blood to coagulate and a higher likelihood of relapse. When the prothrombin time is low, we must infer that the clotting time is too short, which raises the danger of clot formation and may result in disorders like thrombosis. Venous thromboembolism (VTE), which includes deep vein thrombosis and pulmonary embolism, is a frequent consequence of cancer. Cancer increases the chance of VTE by four to seven times, and systemic treatment further increases the risk. Furthermore, cancer-associated VTE is related to worse survival; studies have revealed that VTE is associated with a twofold greater risk of death in cancer patients [48]. On the other hand, the evidence regarding the use of the number of peripheral blood eosinophils as a prognostic parameter in CRC patients is limited; however, there are no convincing data linking blood eosinophilia to prognostic severity in CRC patients. Eosinopenia has been proposed as an independent risk factor for RFS in stage II and III CRC. Nevertheless, no significant association between pretreatment blood eosinophil count and OS in patients with resectable CRC has been documented [49,50].

Several histological response classification systems have been developed for surgical specimens of liver metastases in colorectal tumors following perioperative treatments. This concept, known as tumor regression grade (TRG), was analyzed by Trakarnsanga et al. comparing the concordance indices and prognostic value of six different classifications, concluding that the AJCC/CAP TRG system is the most accurate [51]. Subsequently, Mace et al. suggested that the AJCC/CAP TRG system could be an independent predictor of OS, RFS, and cumulative recurrence. Currently, the National Comprehensive Cancer Network (NCCN) Colorectal Cancer Guidelines Panel recommends AJCC/CAP TRG criteria for pathological response evaluation, but its clinical significance has not yet been fully defined [52]. TRG may represent a useful prognostic factor in patients with mCRC receiving preoperative chemotherapy, although there is no clear evidence about its predictive role [53]. In our project, an expert pathologist reviewed pathological samples after preoperative chemotherapy treatment, establishing a classification system into three ranges (no response, moderate response, and complete response). In the final statistical analysis, it was found that this variable contributed to improving the prediction potential of the statistical model, consistent with the oncological rationale previously studied by other publications.

Ming-ming He et al. demonstrated that levels of ALT or AST within or below the normal range are associated with an increased risk of CRC. Previous studies have reported an inverse association between ALT levels and overall cancer risk or mortality. The EPIC-Heidelberg case-cohort study examined AST in relation to CRC risk and found no associations. These discrepancies, possibly due to the heterogeneity and different methodologies of the studies, prevent liver enzymes from being considered a standardized and reproducible biomarker [54]. Alternatively, while several studies indicate a negative association between pre-operative thrombocytosis and survival in colorectal cancer patients, no definitive consensus exists on this matter. Finally, some studies found that mean platelet volume (MPV) is correlated with patient survival and is an independent risk factor for prognosis, although the underlying mechanisms for the association are not fully understood [55].

Performance status serves as a comprehensive assessment of a patient’s overall functioning and is routinely evaluated in clinical practice. In CRC, it is well established that patients with an ECOG score of zero exhibit better OS, RFS, and a lower likelihood of experiencing severe or fatal adverse events compared to those with an ECOG score of one. The patient’s general health following neoadjuvant therapy (ECOGT) is also an essential factor that influences quality of life, toxicity, and adherence to treatment, with a multifactorial impact on efficacy metrics. In this regard, it is reasonable to expect that an ECOGT condition other than zero correlates with a worse prognosis. 

We attempted to design an algorithm that solely incorporates neoadjuvant factors, as it might be more suitable for routine clinical practice. However, the challenges in analyzing and ensuring data quality are more intricate, making this prediction more challenging when restricting the investigation to the neoadjuvant scenario. Nevertheless, the goal is to make it more pragmatic for patients and oncologists to establish treatment and monitoring decisions early on. Given that surgical intervention remains the mainstay of the current clinical management, the model aims to refine at-risk patients, adapting potential adjuvant therapies according to the predicted risk of relapse, or fostering the development of clinical trials in patients at higher risk.

This report has several limitations. Firstly, the analysis was based on a relatively small number of cases, and therefore the conclusions should be verified through large-scale multicenter clinical studies. Secondly, the retrospective design of this study may not fully account for certain confounding factors and could introduce a degree of bias. Thirdly, there is some heterogeneity in the chemotherapy regimens employed, and the study represents a single institutional experience. Fourthly, the radiological assessment measured by RECIST criteria (version 1.1) and the depth of response were evaluated along with the rest of the predictive model, although this information was not included in the model due to variations in the use of radiological techniques (CT vs hepatic MRI) across patients and the lack of uniformity in the frequency of evaluations during consistent follow-up timeframes. Fifthly, while the precision and sensitivity results of the internal validation were satisfactory, the specificity was lower than expected. Therefore, it is necessary to expand the validation process, particularly by testing a larger number of patients without relapse, to reduce the likelihood of false negatives and thus enhance specificity. Finally, some of the variables found to be statistically significant for the model (e.g., minimal creatinine, eosinophil differential estimator, etc.) currently lack a clinical correlate. In this context, our study is based on analyzing all variables as equal parameters, regardless of their clinical significance. The aim is to enhance the methodology’s robustness and the solidity of the results while avoiding the introduction of clinical associations that might potentially impact the final model.

Despite these limitations, the study also possesses some strengths. It incorporates over 150 variables into the statistical algorithms, allowing for a comprehensive analysis. Additionally, the long-term follow-up of the patients provides valuable insights. Furthermore, the development of an internal validation model with patients having similar characteristics, albeit with a small sample size, exhibits promising predictive behavior. The additional data from the conducted external validation provide greater robustness to the prediction model, making its widespread use in other settings more feasible. The exclusive focus on neoadjuvant treatment facilitates early decision-making for patient management. With the aim of determining the individual relapse risk for each patient, the objective is to acquire a probabilistic estimation for clinical utilization and assessment by healthcare clinicians. In this regard, we are developing a tool that enables physicians to visualize these variables within the model and calculate the individual risk of relapse as a percentage. However, to solidify the findings, larger, multicenter studies are warranted.

## Figures and Tables

**Figure 1 biomedicines-12-01859-f001:**
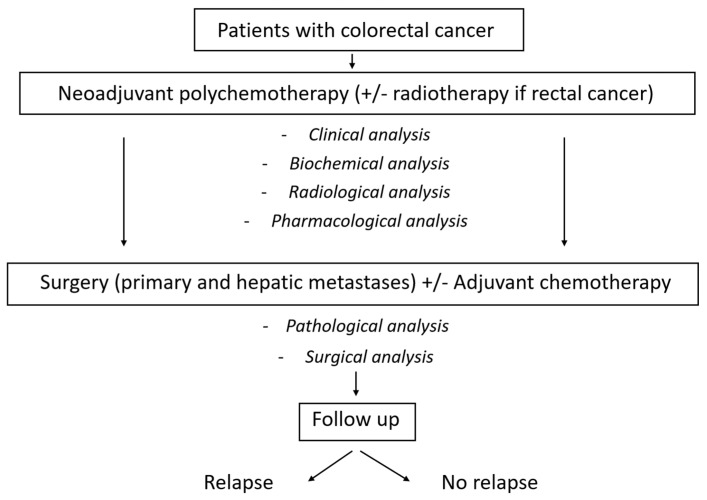
Patient flowchart through the therapeutic algorithm.

**Figure 2 biomedicines-12-01859-f002:**
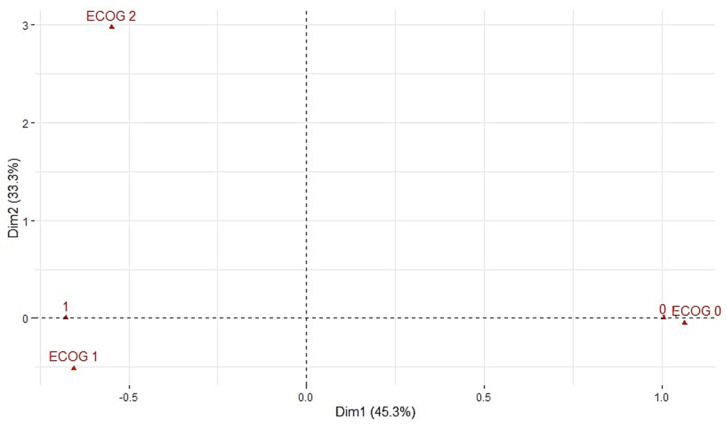
The correspondence analysis graph shows the relationship between ECOGT variable (ECOG scale after neoadjuvant therapy) and the relapse variable. ECOGT categories are represented by points ECOG 0,1,2 while the relapse categories are represented by points 0,1. Two dimensions or variables are used for their interpretation, explaining their relationship by 78.617% (Dimension 1: 45.284; Dimension 2: 33.333).

**Figure 3 biomedicines-12-01859-f003:**
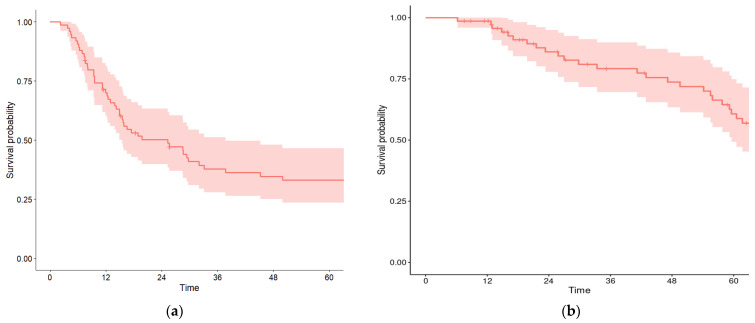
Kaplan–Meier estimates relapse-free survival (**a**) and overall survival (**b**) among patients with potentially resectable LM of CRC treated with perioperative approach. The red line represents the estimated survival probability over time, while the red-shaded area denotes the confidence interval.

**Figure 4 biomedicines-12-01859-f004:**
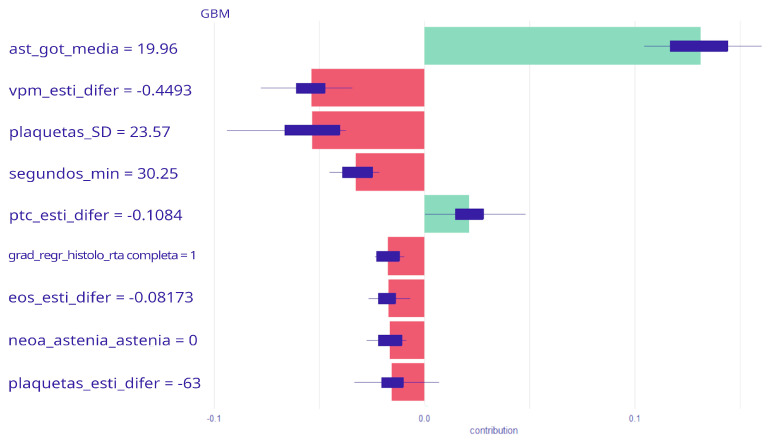
The impact of each feature on the model’s output, with contributions measured in SHAP values. Positive SHAP values indicate that a feature increases the likelihood of a particular prediction, while negative SHAP values suggest a decrease in that likelihood. The features are arranged in descending order of impact. The red bars indicate the negative contribution of the variable to the model’s prediction, reflecting a decrease in the probability of the outcome as the variable’s value changes. The green bars represent a positive contribution, indicating an increase in probability. The blue lines and bars denote the confidence interval or variability in the variable’s contribution.

**Table 1 biomedicines-12-01859-t001:** Baseline characteristics.

	Variables	N (%)
Age-years	Median Range	59 28–79
Gender	Male Female	47 (63.5) 27 (36.5)
ECOG	0 1	59 (79.9) 15 (20.1)
Location	Colon Rectum	53 (71.6) 21 (28.4)
Presentation	Synchronic Metachronic	62 (83.8) 12 (16.2)
Number of LM	≤3 >3	59 (79.2) 15 (20.8)
ESMO CG	Group 0/1	74 (100)
Köhne Group *	Köhne 1	74 (100)
Neoadjuvant chemotherapy	FOLFOXIRI FOLFOXIRIm XELOXIRI FOLFOX XELOX	11 (14.8) 26 (21.6) 29 (39.2) 12 (16.2) 6 (8.1)
Liver surgery	Atypical resection/s Segmentectomy Right hepatectomy Left hepatectomy Central hepatectomy	41 (55.4) 10 (13.5) 17 (23) 3 (4.1) 3 (4.1)
Age-years	Median Range	59 28–79

* Köhne et al. established 3 prognostic groups for patients with stage IV CRC treated with chemotherapy using 4 clinical characteristics (Eastern Cooperative Oncology Group or ECOG, PS, number of metastatic sites, alkaline phosphatase level, and leukocyte count) [41].

**Table 2 biomedicines-12-01859-t002:** Features included in the model.

**Categorical Variables**				
**Variables**	**Categories**			**Values**
Asthenia	No asthenia Asthenia			291 (34%) 559 (66%)
TRG	No response Moderate response Complete response			275 (32%) 259 (30%) 317 (37%)
Origin primary tumor	Colon Rectum			574 (68%) 276 (32%)
ECOGT	ECOG 0 NO-ECOG 0			529 (62%) 321 (38%)
**Non-Categorical Variables**				
**Variables**	**Min**	**Mean**	**Median**	**Max**
Mean of AST	7.35	24.07	20.49	96.77
Platelets SD	1.66	63.68	56.14	182.94
Eosinophils DE	−0.47	−0.065	−0.052	0.34
VPM DE	−2.28	−0.3	−0.25	1.2
Mean of creatinine	0.47	0.88	0.84	2.40
Min sec coagulate blood	22.61	30.09	29.55	43.36
PTC DE	−0.4	−0.084	−0.10	0.1
Min creatinine	0.40	0.78	0.71	1.99
Platelets DE	−471	−92.45	−76	96
Maximum AST	8	33.36	29	140
CHCM DE	−2.59	0.37	0.32	5.1

AST: aspartate aminotransferase, TRG: tumor regression grade, ECOGT: ECOG performance status after neoadjuvant treatment, VPM: mean platelet volume, PTC: prothrombin time concentration, CHCM: mean corpuscular hemoglobin concentration, SD: standard deviation, DE: differential estimator, Min: minimum, sec: seconds.

**Table 3 biomedicines-12-01859-t003:** Internal validation dataset characteristics.

	Variables	N (%)
Age-years	Median Range	62 34–76
Gender	Male Female	15 (65.2) 8 (34.8)
ECOG	0 1	18 (78.3) 5 (21.7)
Location	Colon Rectum	13 (56.5) 10 (43.5)
Presentation	Synchronic Metachronic	18 (78.3) 5 (21.7)
Number of LM	≤3 >3	17 (73.9) 6 (26.1)
ESMO CG	Group 0/1	23 (100)
Köhne Group	Köhne 1	23 (100)
Neoadjuvant chemotherapy	FOLFOXIRI XELOXIRI FOLFOX XELOX	8 (34.8) 6 (26.1) 5 (21.7) 4 (17.4)
Adjuvant Chemotherapy	Yes No	14 (60.9) 9 (39.1)
Liver surgery	Atypical resection/s Segmentectomy Right hepatectomy Left hepatectomy Central hepatectomy	13 (56.6) 3 (13.0) 3 (13.0) 3 (13.0) 1 (4.4)

**Table 4 biomedicines-12-01859-t004:** Basic characteristics of internal validation data set that were included in the predictive model.

**Categorical Variables**				
**Variables**	**Categories**			**Values**
Asthenia	No asthenia Asthenia			19 (83%) 4 (17%)
TRG	No response Moderate response Complete response			7 (30%) 0 (0%) 16 (70%)
Origin primary tumor	Colon Rectum			13 (57%) 10 (43%)
ECOGT	ECOG 0 NO-ECOG 0			2 (9%) 21 (91%)
**Non-Categorical Variables**				
**Variables**	**Min**	**Mean**	**Median**	**Max**
Mean of AST	10.97	19.84	19.09	41.85
Platelets SD	7.41	46.48	41.51	134.92
Eosinophils DE	−0.33	−0.029	−0.02	0.16
VPM DE	−2.07	−0.27	−0.20	1.4
Mean of creatinine	20.1	30.1	28.25	43.10
Min sec coagulate blood	0.55	0.86	0.85	1.22
PTC DE	−0.3	−0.034	0	0.1
Min creatinine	0.47	0.76	0.79	1.11
Platelets DE	−386	−36.74	−23	144
Maximum AST	12	46.58	29	249
CHCM DE	−2	−0.26	−0.35	1

AST: aspartate aminotransferase, TRG: tumor regression grade, ECOGT: ECOG performance status after neoadjuvant treatment, VPM: mean platelet volume, PTC: prothrombin time concentration, CHCM: mean corpuscular hemoglobin concentration, SD: standard deviation, DE: differential estimator, Min: minimum, sec: seconds.

**Table 5 biomedicines-12-01859-t005:** Confusion matrix of internal validation.

**Predict**		**Real**	
		NO	YES
	NO	1	2
	YES	3	17

**Table 6 biomedicines-12-01859-t006:** Basic characteristics of external validation data set that were included in the predictive model.

**Categorical Variables**				
**Variables**	**Categories**			**Values**
Asthenia	No Asthenia Asthenia			17 (71%) 7 (29%)
TRG	No response Moderate response Complete response			8 (33%) 10 (42%) 6 (25%)
Origin primary tumor	Colon Rectum			18 (75%) 6 (25%)
ECOGT	ECOG 0 NO-ECOG 0			5 (21%) 19 (79%)
**Non-Categorical Variables**				
**Variables**	**Min**	**Mean**	**Median**	**Max**
Mean of AST	12.78	26.88	26.31	46.69
Platelets SD	9.64	48.18	39.66	106.01
Eosinophils DE	−0.61	−0.093	−0.075	0.11
VPM DE	−2	−0.075	−0.1	2.6
Mean of creatinine	23.4	31.75	31.3	40
Min sec coagulate blood	0.52	0.89	0.84	2.18
PTC DE	−0.25	−0.048	−0.06	0.13
Min creatinine	0.4	0.77	0.75	1.7
Platelets DE	−275	−44	−53	128
Maximum AST	18	43.37	39	88.9
CHCM DE	−1.4	−0.038	0	2

AST: aspartate aminotransferase, TRG: tumor regression grade, ECOGT: ECOG performance status after neoadjuvant treatment, VPM: mean platelet volume, PTC: prothrombin time concentration, CHCM: mean corpuscular hemoglobin concentration, SD: standard deviation, DE: differential estimator, Min: minimum, sec: seconds.

**Table 7 biomedicines-12-01859-t007:** Confusion matrix of external validation.

**Predict**		**Real**	
		NO	YES
	NO	3	3
	YES	3	14

## Data Availability

No new data were created or analyzed in this study. Data sharing is not applicable to this article.

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
