# Peer review of "Neoadjuvant Statistical Algorithm to Predict Individual Risk of Relapse in Patients with Resected Liver Metastases from Colorectal Cancer"

_biomedicines, 2024, doi:10.3390/biomedicines12081859_

Round 1
Reviewer 1 Report
Comments and Suggestions for Authors
Atienza et al. propose a statistical algorithm for the risk assessment in the case of Neoadjuvant in patients with resected liver metastases from colorectal cancer.
Given the complex nature of cancer, prediction of risk in case of surgical outcome is always a matter of concern in cancer patients including colorectal cancer. Therefore, statistical algorithms, uses of AI, and machine learning tools are always helpful in risk assessment.
Here are some suggestions that need to be discussed and elaborated.
1. Predictive model for relapse risk in colorectal cancer should be highlighted in the background of genetic heterogeneity among the selected cancer patients.
2. The authors may address the association of the timing of chemotherapy and surgery among various colorectal cancer patients and release risk prediction.
3. Since the size of the selected patient is not so appreciable what are the alternative options that can be discussed?
4. The authors should have tested with an alternative or known statistical package and validated with the proposed algorithm.
5. The authors should highlight the future scope of uses of the algorithm in case of occurrence of two primary tumors and their metastatic tumors.
Comments on the Quality of English Language
Moderate
Author Response
Attached you will find a point-by-point response to all the suggestions raised by the reviewers. We would like to express our gratitude for the in-depth revisions provided, as we believe that the quality of the manuscript has significantly improved thanks to these suggestions. We hope that the manuscript meets the quality standards of the journal "Biomedicines." Please let us know if you need any further clarification.
Best regards,
Ángel Vizcay Atienza
Atienza et al. propose a statistical algorithm for the risk assessment in the case of Neoadjuvant in patients with resected liver metastases from colorectal cancer.
Given the complex nature of cancer, prediction of risk in case of surgical outcome is always a matter of concern in cancer patients including colorectal cancer. Therefore, statistical algorithms, uses of AI, and machine learning tools are always helpful in risk assessment.
Here are some suggestions that need to be discussed and elaborated.
- Predictive model for relapse risk in colorectal cancer should be highlighted in the background of genetic heterogeneity among the selected cancer patients.
We completely agree; the genetic heterogeneity of colorectal cancer is one of the major challenges in homogenizing patient groups due to its inherent molecular complexity. We have included this consideration as one of the limitations in the evaluation of certain prognostic groups of colorectal cancer within our algorithm (lines 494-497)
- The authors may address the association of the timing of chemotherapy and surgery among various colorectal cancer patients and release risk prediction.
The specified time interval between chemotherapy and surgery refers to the period from the initiation of chemotherapy to the surgical intervention. This variability depends on the number of cycles administered to each patient, which can vary based on the individualized objective to achieve an optimal response prior to surgery (lines 252-254). We have also included the 5-year median survival data for patients in this scenario (lines 55-57) and its references.
- Since the size of the selected patient is not so appreciable what are the alternative options that can be discussed? We have utilized and attempted to convey the concept of synthetic data (2.4.4) and GBM (2.4.3) to optimize this limitation. The future goal is to increase the number of recruited patients. In this regard, thanks to previously published data on synthetic data, they allow for increasing the value and number of scientific data by generating simulated individuals who share similar characteristics in an anonymized manner, thereby accelerating precision medicine studies and the development of clinical trials. We have emphasized this idea in lines 514-516.
- The authors should have tested with an alternative or known statistical package and validated with the proposed algorithm.
We appreciate this comment because it was one of the major methodological changes in our work. We have retained section 2.4.2, explaining the conventional statistical method of logistic regression, which was used to analyze the 74 valid patients, yielding good results with this method: accuracy, sensitivity, and specificity scores (0.80, 0.84, and 0.83, respectively) for predicting relapse risk. Therefore, although we have retained the initial concepts of logistic regression in the article, we subsequently focused on GBM and synthetic data with the aim of introducing new concepts and attempting to improve the limitation of the number of patients.
- The authors should highlight the future scope of uses of the algorithm in case of occurrence of two primary tumors and their metastatic tumors.
We appreciate this observation; however, the synchronous presentation of two colorectal tumors is low, estimated at around 3%. Nevertheless, this situation would not be included in our prediction model, as it does not meet the original selection criteria. We have highlighted the applicability of the model under the original conditions established in our population in lines 502-505, considering this other cohort a good idea for designing future studies.

Reviewer 2 Report
Comments and Suggestions for Authors
The paper by Viscay Atienza et al is describing the development of an interpretable neoadjuvant algorithm based on mathematical models to accurately predict individual risk, ensuring mathematical transparency and auditability for liver metastases of colorectal cancer (CRC). 86 patients with liver metastases treated with neoadjuvant systemic therapy and complete surgical resection were retrospectively evaluated. In addition, analysis of 155 individual patient variables was performed. A predictive model for relapse risk through significance testing and ANOVA analysis was performed by using logistic regression.
The authors conclude that their model offers an alternative for predicting individual relapse risk, aiding in personalized adjuvant therapy and follow-up strategies.
The paper is clearly written, and the language is good.
Here are some points that need to be considered:
1. Please explain the term ECOG.
2. Which “original” database was used (as described in section 3.1.4. Synthetic patient data)? Please give a more extensive description in this section.
3. I would prefer a longer introduction including some of the variables described in Table 2, giving some more background to the importance of these variables in mCRC.
4. I agree with the authors that the low number of cases is a limitation. Hopefully, this can be adjusted in a future study.
Comments on the Quality of English LanguageMinor editing of English language required
Author Response
Attached you will find a point-by-point response to all the suggestions raised by the reviewers. We would like to express our gratitude for the in-depth revisions provided, as we believe that the quality of the manuscript has significantly improved thanks to these suggestions. We hope that the manuscript meets the quality standards of the journal "Biomedicines." Please let us know if you need any further clarification.
Best regards,
Ángel Vizcay Atienza
The paper by Vizcay Atienza et al is describing the development of an interpretable neoadjuvant algorithm based on mathematical models to accurately predict individual risk, ensuring mathematical transparency and auditability for liver metastases of colorectal cancer (CRC). 86 patients with liver metastases treated with neoadjuvant systemic therapy and complete surgical resection were retrospectively evaluated. In addition, analysis of 155 individual patient variables was performed. A predictive model for relapse risk through significance testing and ANOVA analysis was performed by using logistic regression.
The authors conclude that their model offers an alternative for predicting individual relapse risk, aiding in personalized adjuvant therapy and follow-up strategies.
The paper is clearly written, and the language is good.
Here are some points that need to be considered:
- Please explain the term ECOG. The definition of ECOG was added in line 107. The explanation of the different values was also added to the text of Figure 2.
- Which “original” database was used (as described in section 3.1.4. Synthetic patient data)? Please give a more extensive description in this section. We refer to the original database as the initial one, which included the electronic health records of the 74 patients, upon which synthetic data was subsequently applied. We have completed the information in line 282.
- I would prefer a longer introduction including some of the variables described in Table 2, giving some more background to the importance of these variables in mCRC. As we outlined in the methodology, approximately 150 variables were collected from each patient. The foundation of the algorithm is to treat these clinical and analytical variables equally, aiming to obtain results from the interaction and training of mathematical models. In this regard, the variables in Table 2 were not previously anticipated, so the explanation of their subsequent importance is presented in the discussion section rather than in the introduction. We hope that this explanation of the methodology behind our algorithm's development aligns with your perspective on the interpretation of the variables presented in the results section of the conclusions.
- I agree with the authors that the low number of cases is a limitation. Hopefully, this can be adjusted in a future study. We completely agree; it is one of our main goals to strengthen our data. In this vein, we have employed novel techniques of synthetic data and its evaluation by GBM with the aim of addressing this limitation.
